Machine learning algorithms accurately identify free-living marine nematode species

Brito de Jesus Simone 1 simone.brito@unifesp.br
Vieira Danilo 1
Gheller Paula 2
Cunha Beatriz P. 3
Gallucci Fabiane 1
Fonseca Gustavo 1
1 Marine Science Institute, Federal University of São Paulo , Santos, São Paulo , Brazil
2 Institute Oceanographic, University of São Paulo , São Paulo , Brazil
3 Department of Animal Biology, State University of Campinas , Campinas, São Paulo , Brazil
Waiho Khor
Electronic publication date: 2023 Oct 9
Publication date: 2023
Volume: 11
Electronic Location ID: e16216
Received 2023 Jun 21; Accepted 2023 Sep 11
Copyright: © 2023 Brito de Jesus et al.
Copyright year: 2023
Copyright holder: Brito de Jesus et al.
License: This is an open access article distributed under the terms of the Creative Commons Attribution License, which permits unrestricted use, distribution, reproduction and adaptation in any medium and for any purpose provided that it is properly attributed. For attribution, the original author(s), title, publication source (PeerJ) and either DOI or URL of the article must be cited.
License URL: https://creativecommons.org/licenses/by/4.0/

Keywords: Nematoda, Identification-key, Acantholaimus, Sabatieria, Random Forest, Support vector machine, Stochastic gradient boosting, K-nearest neighbor

Funding: Conselho Nacional de Desenvolvimento Científico e Tecnológico—CNPQ to Gustavo Fonseca 306780/2022-4 Financial support was provided by the Conselho Nacional de Desenvolvimento Científico e Tecnológico—CNPQ to Gustavo Fonseca (306780/2022-4). The funders had no role in study design, data collection and analysis, decision to publish, or preparation of the manuscript.

==============================
Background

Identifying species, particularly small metazoans, remains a daunting challenge and the phylum Nematoda is no exception. Typically, nematode species are differentiated based on morphometry and the presence or absence of certain characters. However, recent advances in artificial intelligence, particularly machine learning (ML) algorithms, offer promising solutions for automating species identification, mostly in taxonomically complex groups. By training ML models with extensive datasets of accurately identified specimens, the models can learn to recognize patterns in nematodes’ morphological and morphometric features. This enables them to make precise identifications of newly encountered individuals. Implementing ML algorithms can improve the speed and accuracy of species identification and allow researchers to efficiently process vast amounts of data. Furthermore, it empowers non-taxonomists to make reliable identifications. The objective of this study is to evaluate the performance of ML algorithms in identifying species of free-living marine nematodes, focusing on two well-known genera: Acantholaimus Allgén, 1933 and Sabatieria Rouville, 1903.

Methods

A total of 40 species of Acantholaimus and 60 species of Sabatieria were considered. The measurements and identifications were obtained from the original publications of species for both genera, this compilation included information regarding the presence or absence of specific characters, as well as morphometric data. To assess the performance of the species identification four ML algorithms were employed: Random Forest (RF), Stochastic Gradient Boosting (SGBoost), Support Vector Machine (SVM) with both linear and radial kernels, and K-nearest neighbor (KNN) algorithms.

Results

For both genera, the random forest (RF) algorithm demonstrated the highest accuracy in correctly classifying specimens into their respective species, achieving an accuracy rate of 93% for Acantholaimus and 100% for Sabatieria, only a single individual from Acantholaimus of the test data was misclassified.

Conclusion

These results highlight the overall effectiveness of ML algorithms in species identification. Moreover, it demonstrates that the identification of marine nematodes can be automated, optimizing biodiversity and ecological studies, as well as turning species identification more accessible, efficient, and scalable. Ultimately it will contribute to our understanding and conservation of biodiversity.

Introduction

The correct taxonomic identification of species forms the foundation for biodiversity, ecology, phylogeny, and conservation studies. Traditionally, species identification has relied on the use of dichotomous keys based on morphological characters (Griffing, 2001; De & Dey, 2019). Despite the advent of DNA barcoding, morphological identification remains prevalent, primarily due to the limitations of DNA reference databases (Blaxter, 2004; Valentini, Pompanon & Taberlet, 2009; Guo et al., 2022). However, dichotomous keys are often limited to a specific geographic area, a small number of species, and a restricted set of morphological characters (Osborne, 1963; Walter & Winterton, 2007). Alternative tools such as polytomous keys (Weiss, 1995), pictorial keys (Schmidt-Rhaesa, 2014), and tabular keys (Fonseca, Vanreusel & Decraemer, 2006) have been proposed but also show similar limitations. To address these challenges, studies have explored the use of multivariate statistical techniques to analyze various morphological characteristics and morphometric measures simultaneously (Bailey & Byrnes, 1990; Stock & Kaya, 1996; Shokoohi & Moyo, 2022). While these approaches have been useful in grouping similar specimens and providing a more objective basis for species delimitation, their effectiveness in identifying new individuals, as expected from an identification key, has not been adequately evaluated. Thus, the challenge of evaluating newly collected specimens and assigning appropriate species names remains, hindering progress in research reliant on species identification.

In recent years, machine learning (ML) algorithms have emerged as a powerful tool to enhance data processing and facilitate species identification across taxa, including birds, insects, and plants (Wäldchen & Mäder, 2018; Islam et al., 2019; Kasinathan, Singaraju & Uyyala, 2021; Bojamma & Shastry, 2021). The fundamental principle behind ML-based species identification involves leveraging existing taxonomic knowledge, where each new observation is assigned a probability of belonging to a previously described species. Notably, a common aspect of these ML studies is that the identification was done on images or, in the case of birds, their songs and calls as well (Jadhav, Patil & Parasar, 2020; Mehyadin et al., 2021). Nonetheless, the application of ML approaches is not limited to images or audio data but can be extended to virtually any data type. This is particularly relevant in cases where obtaining high-quality images is challenging or not always possible. In such instances, species identification often relies on numerical data matrices that combine morphometric measurements and the presence/absence of morphological characters (Larrazabal-Filho, Neres & Esteves, 2018; Maria et al., 2009; Surmacz, Morek & Michalczyk, 2020; Tumanov, 2020; Mitra et al., 2019). In this regard, machine learning techniques can also potentially be effectively utilized for species identification. Supervised algorithms can be employed in these cases to automate the identification process. These algorithms utilize the species labels as the supervised variable (Y) and the morphological characteristics as the predictors (X). By training the algorithm on this data, it can learn the patterns and relationships between the morphological features and the corresponding species.

The aim of this study is to evaluate the performance of multiple machine learning algorithms on the identification of free-living marine nematode species. Free-living marine nematodes are small invertebrates that belong to the meiofauna. They are highly abundant and species-rich (Vanreusel, Fonseca & Danovaro, 2010; Hauquier et al., 2019; Zeppilli et al., 2019). These organisms are known as good ecological indicators due to their ubiquitous presence in diverse ecosystems and sensitivity to environmental changes (Moreno et al., 2011; Bianchelli et al., 2018). Moreover, they play a crucial role in various ecosystem functions, such as mineralization, oxygenation of the sediment, and secondary productivity (Schratzberger & Ingels, 2018).

Despite their ecological importance, the lack of reliable identification tools at lower taxonomical levels hampers ecological, molecular, and conservation studies (Macheriotou et al., 2019; Ridall & Ingels, 2021; Pantó et al., 2021). As a result, nematodes are often identified at the genus level rather than the species level (Miljutin et al., 2010; Sandulli, Semprucci & Balsamo, 2014; Brannock et al., 2017; Spedicato et al., 2020). The use of ML techniques in nematode identification is still limited. It has been successfully applied in the identification of species through image analysis (Thevenoux et al., 2021) and in the processes of detecting morphological and phenotypic features (Hakim et al., 2018). Although incipient, the initiatives demonstrated the versatility and potential of using machine learning in nematodes. The methodology proposed in this study will be tested using individuals from the genera Acantholaimus and Sabatieria. Acantholaimus (Allgén, 1933) is typically found in the deep sea (Miljutin & Miljutina, 2016a). Sabatieria (Rouville, 1903) is one of the most abundant and dominant genera along continental shelves and slopes, serving as an indicator of ecosystem wealth (Vanreusel, Fonseca & Danovaro, 2010; Kotwicki, Grzelak & Bełdowski, 2016; Mincks et al., 2021). Both genera are characterized by a large morphological variation, the presence of many described species, and have recent taxonomic reviews (Miljutin & Miljutina, 2016a; Venekey et al., 2019; Fonseca & Bezerra, 2014; Yang et al., 2019) making them highly suitable for testing ML tools for species identification.

Material and Methods

Literature review

The first step towards testing ML algorithms in the identification of Acantholaimus and Sabatieria species was to list all valid species described for each genus. All taxonomic descriptions and reviews considering these two genera were used in this study (Tables S1 and S2). Species for which publication provided the measurements of a single individual or descriptions that lacked significant taxonomic information were not included in the analysis. Considering these criteria, for Acantholaimus, a total of 40 out of the 46 valid species were considered (Table S1), while for Sabatieria, a total of 60 out of the 107 species were included (24 species were excluded due to the absence of information of characters and 23 were excluded because the description was limited to a single specimen; Table S2). Below we present a brief description of the genera and the morphological characters used for species identification in this study. To describe each species, body regions are abbreviated using the De Man (1880) and Cobb (1917) system of indices.

Morphological and morphometric data

Acantholaimus

The genus Acantholaimus Allgén, 1933 belongs to the family Chromadoridae, Filipjev, 1917, subfamily Spilipherinae, and it includes 46 valid species (Venekey et al., 2019; Holovachov, 2020; Manoel, Esteves & Neres, 2022). Venekey et al. (2019) provided the latest diagnosis of the genus.

The selection of characters to be included in the model was based on De Mesel et al. (2006) and Miljutin & Miljutina (2016b): 14 morphometric measurements (in µm); eight quantitative ratios; and seven categorical morphological characters, namely the amphid position (AP), amphid size (AS), cervical setae position (CSP), head shape (HS), pharynx shape (Pha.S), cuticular ornamentation (CO) and tail shape (TS). All morphometric characters for both genera are depicted in Table 1. For each morphological character, categories were assigned as detailed below: Head shape (HS): Acantholaimus species may have one out of four different head shapes: (a) truncated; (b) round; (c) tapered; and (d) narrow (Fig. 1A).

Cervical setae position (CSP): In general, a pair of cervical setae is located posterior to the base of the amphid in each sublateral line, but the distance from the posterior border of the amphid varies between species. Three categories were established (Fig. 1B): (a): anterior or at the level of the posterior border of the amphid; (b): moderate distance in relation to the posterior border of the amphid (<1.0 AH); and (c): distant from the posterior border of the amphid (=or >1.0 AH).

Amphid size (AS): The amphid size was estimated considering the ratio between its height (AH) and the corresponding body diameter (CD). Four categories were established (Fig. 1C): (a) AH/CD ≈ 1 (very large); (b): 1 > AH/CD > 0.5 (large); (c): AH/CD ≈ 0.5 (medium); and (d): AH/CD < 0.5 (small).

Amphid position (AP): The amphid position was assessed considering the ratio between the distance from the anterior end to the amphid anterior borderline (AAE) and the corresponding body diameter at the mid-amphideal level (CD). Three categories were separated (Fig. 1D): (a): <1.0 (close to head end); (b): ≈1.0 (mid-amphideal); and (c): >1.0 (behind anterior end).

Pharynx shape (Pha.S): Often, the pharynx is thick and muscular with numerous plasmatic interruptions. Three categories were assigned: (a): cylindrical; (b): round bulb; and (c): elongated bulb (Fig. 1E).

Cuticular ornamentation (CO): The cuticle is ornamented with transverse rows of numerous punctuations. The lateral field of the cuticle may be distinguished by a wide lateral differentiation comprising larger, more sparsely, and sometimes more irregularly distributed punctations, or by several longitudinal rows of bigger dots. Three categories were assigned: (a): cuticle without lateral differentiation; (b): lateral differentiation of larger dots arranged irregularly; and (c): lateral differentiation of larger dots arranged in longitudinal rows (Fig. 1F).

Tail shape (TS): Usually, the tail of the Acantholaimus species is conical-cylindrical and long. The change from conical to cylindrical can be abrupt, with a proximal conical section distinct from a distal filiform cylindrical section or gradually tapered to the tip. Two categories were established: (a): tail conical-cylindrical with the distinct filiform part distal section; and (b): tail with proximal conical section gradually tapered, and elongated, smoothly transitioning to the filiform distal section (Fig. 1G).

Table 1 List of selected morphometric characters used for the identification of Acantholaimus and Sabatieria species.

Code	Measurement	Acantholaimus	Sabatieria	
L	Total body length (µm)	✓	✓	
L′	Body length without tail	✓		
Amphid D	Amphid diameter	✓	✓	
OLSL	Length of outer labial setae	✓		
CSL	Length of cephalic setae	✓	✓	
Cerv. LS	Length of cervical setae	✓		
SSL	Length of somatic setae	✓		
Spic.arc	Length of spicule in the arc	✓	✓	
D.A.E. A	Distance from anterior end to amphid	✓		
D.L.C. S	Diameter at the level of cephalic setae	✓		
D.L.M. A	Diameter at the level of the middle of the amphid	✓		
D.L.C	Diameter at the level of cardia	✓		
D.L. A	Diameter at the level of anus	✓		
MBD	Maximum body diameter	✓		
HD	Head diameter		✓	
B.C. W	Buccal cavity width		✓	
Amphid. H	Amphid height		✓	
Amphid. AE	Amphid from the anterior end		✓	
Nerv.ring	Nerve ring from the anterior end		✓	
Pha.L	Pharynx length		✓	
Pha.BBD	Pharyngeal bulb body diameter		✓	
Gub.apoph. L	Gubernacular apophyses length		✓	
Suppl. N°	Number of supplements		✓	
abd	Anal body diameter		✓	
TL	Tail length		✓	
TL/abd	Tail length abd		✓	
a, b, c	De Man’s ratios	✓	✓	
a′, b′, c′	De Man’s ratios	✓	✓	
V	Distance from anterior end to vulva/total body length %	✓	✓	
V′	Distance from anterior end to vulva/body length without tail %	✓	✓	

Figure 1 Morphological characters and diagnostic categories considered for Acantholaimus species.

Sabatieria

Sabatieria (Rouville, 1903) belongs to the family Comesomatidae (Filipjev, 1918), within the subfamily Sabatieriinae (Filipev, 1934). This genus is relatively speciose with 107 valid species (Fu, Leduc & Zhao, 2019; Yang et al., 2019; Zhai, Wang & Huang, 2020; Leduc & Zhao, 2023). The latest diagnosis has been presented by Fonseca & Bezerra (2014).

According to the literature survey, 16 measurements (in µm); six quantitative ratios (Table 1), and eight categorical morphological characters were selected to characterize the species of this genus. The categorical variables were buccal cavity (BC), number of amphideal turns (Amphid. Turn), cuticular ornamentation (CO), spicules (Spic), apophyses shape (Apoph), supplements aspect (Suppl. A), supplements position (Suppl. P), and tail shape (TS). The categories of each morphological character are as follows: Buccal cavity (BC): Within the genus Sabatieria, the degree of cuticularization of the buccal cavity is an important feature to distinguish the species (Jensen, 1979). Three categories were assigned: (a): without cuticularization, where the small buccal cavity is cup-shaped and narrow in the posterior portion; (b): little cuticularization, where the cup-shaped buccal cavity is slightly cuticularized at the base; and (c): with cuticularization, where the cup-shaped buccal cavity has a cuticularized like a tooth (Fig. 2A).

Number of amphideal turns (Amphid. Turn): The genus Sabatieria has a spiral amphid fovea with usually 2 to 3 turns. The number of spiral turns and the percentage of the amphid fovea diameter (compared to the corresponding body diameter) have intraspecific variations (Platt, 1985). For the amphideal fovea number of turns, three categories were chosen: (a): 2–2.5 spiral turns; (b): 3–3.5 spiral turns, and (c): 4–4.5 spiral turns (Fig. 2B).

Cuticular ornamentation (CO): This genus has a punctuated cuticle with or without lateral differentiation of larger punctations regularly or irregularly arranged. For the ornamentation of the cuticle, three categories were chosen: (a) without lateral differentiation; (b) lateral differentiation of larger and irregularly arranged punctations; and (c) lateral differentiation of larger and regularly arranged punctations (Fig. 2C).

Supplements aspect (Suppl. A): The precloacal supplement aspect is also relevant for species delimitation within Sabatieria. The character was classified into three categories: (a) pore-like or tubular; (b) papillae; and (c) not visible when there is no display of that character (Fig. 2D).

Supplements position (Suppl. P): For the distribution pattern of the precloacal supplements, three categories were designated: (a) uniform, when the spacing between the supplements is equal; (b) anterior closer, when the spacing between supplements increases toward the posterior part of the body; and (c) posterior closer, when the spacing between supplements decreases toward the posterior part of the body (Fig. 2E).

Spicules (Spic): The size of the spicule is an essential characteristic of the differentiation of Sabatieria species. The character was classified into three categories considering the relation of the spicules length (SL) by the anal body diameter (ABD): (a) short, with SL/ABD < 1.0–1.3; (b) medium, with SL/ABD ≈ 1.3–1.6; and (c) long, with SL/ABD > 1.6 (Fig. 2F)

Tail shape (TS): Most species of Sabatieria have a conical-cylindrical tail, consisting of an anterior conical portion and a posterior cylindrical portion with a drop-shaped tail tip and three short terminal setae. However, there are species with a conical (blunt) tail, and the lengths between the conical and the cylindrical portion are different, being an important characteristic to differentiate the species. Four categories were assigned: (a) conical, short tail with a rounded or blunt distal portion; (b) short conical-cylindrical, cylindrical distal portion with a length less than a conical anterior portion and slightly clavate tip; (c) medium conical-cylindrical, distal cylindrical portion similar in length to the conical anterior portion; and (d) long conical-cylindrical, cylindrical distal portion longer than the conical anterior portion (Fig. 2G).

Apophyses shape (Apoph): The males of Sabatieria species usually present gubernaculum provided with apophyses that may have three different formats: (a) straight; (b) curved; and (c) complex (with loops or more than one curve) (Fig. 2H).

Figure 2 Morphological characters and diagnostic categories considered for Sabatieria species.

Data analysis

Pre-process data

Encoding categorical data

Prior to the analysis, categorical morphological characters were transformed into numeric variables using two techniques: Integer Encoding and One-Hot Encoding (Dahouda & Joe, 2021; Fig. 3). The criteria for choosing the appropriate encoding technique for each categorical variable were based on the domain knowledge and understanding of the data, as well as the characteristics of the variables themselves. This involves distinguishing between nominal features which have a binary nature from those that have an ordinal nature. By using the most suitable encoding method for each type of categorical data, we aimed to optimize the representation of the information and enhance the model’s ability to learn and make accurate predictions. The integer encoding technique assigned a unique integer value to each category, with a fixed reference level. They are used for categorical variables with ordinal relationships, where the categories have a specific order or hierarchy. For Acantholaimus, morphological characters such as amphid position (AP), amphid size (AS), and cervical setae position (CSP) were encoded as integers. One-Hot Encoding transformed each variable with n observations and d distinct values into d binary variables with n observations. Each observation indicated the dichotomous binary variable’s presence (1) or absence (0). For Acantholaimus, characters such as head shape (HD), pharynx shape (Pha.S), cuticular ornamentation (CO), and tail shape (TS) were treated as binary. For Sabatieria, morphological characters such as the number of amphideal turns (Amphid. Turn), spicules (Spic), apophyses shape (Apoph), and were encoded as integers. Characters like buccal cavity (BC), supplements aspect (Suppl. A), supplements position (Suppl. P), cuticular ornamentation (CO), and tail shape (TS) were treated as binary variables.

Figure 3 The workflow for applying machine learning algorithms.

Dataset acquisition, data analysis and output. The chosen dataset, sourced from descriptions literature on Acantholaimus and Sabatieria species, was organized into matrix labels representing individuals and their corresponding morphological and morphometric characteristics. This organized data served as the input for the subsequent machine-learning stages. The selection and classification algorithms employed encompassed Random Forest, Stochastic Gradient Boosting, Support Vector Machine, and K-nearest neighbor techniques. These algorithms were utilized to identify the optimal set of features for species recognition and to construct predictive models for accurately identifying individuals based on the presence/absence of morphological and morphometric characteristics.

Handling the missing data and feature scaling

Data imputation was performed to address missing values in some morphometric characters of both genera. To ensure a conservative analysis and avoid potential bias, imputation was done by replacing missing values with the mean value of the respective character across the genus. Additionally, the data was scaled before applying the algorithms. Scaling was necessary to ensure fair comparisons, accurate distance calculations, and reliable predictions (Sukumar, 2014). It also helps to eliminate biases introduced by varying scales and enhances the algorithm’s performance.

Splitting the dataset

To validate the identification of the two models constructed for each genus, the input data for all the algorithms were split into training and testing sets. The minimum number of individuals required to perform the split is four (one for testing and three to perform the cross-validation in the training data). So, only species of Acantholaimus and Sabatieria which were described based on four or more individuals were included in the testing set (Supplemental Material Tables S3 and S4). For descriptions based on 4–9 individuals, one was randomly left out for validation, whereas for descriptions based on more than 10 individuals, two individuals were randomly left out. For the Acantholaimus model, the training set had 131 individuals from the 40 species, and the testing set had 14, resulting in a total of 145 individuals. In the case of the Sabatieria model, out of the 60 species, 227 individuals were used for training and 33 individuals were used for testing, totaling 260 individuals.

Machine-learning analysis

Algorithms

Four algorithms were selected to generate the identification models for Acantholaimus and Sabatieria species: Random Forest (RF), Stochastic Gradient Boosting (SGboost), Support Vector Machine (SVM; linear and radial), and K-nearest neighbor (KNN). The RF algorithm consists of a set of decision trees generated within the same object. Each object, which consists of multiple trees, undergoes a voting mechanism (bagging) to determine the most voted classification (Knauer et al., 2019; Shaik & Srinivasan, 2019). SGBoost combines simple decision trees, known as weak models (Hastie, Tibshirani & Friedman, 2001), to create a strong classifier (Natekin & Knoll, 2013). The SVM (linear and radial) method is a popular classification algorithm that plots each sample data in an n-dimensional space, where n is the number of features. The SVM algorithm then finds the best-fit hyperplane that maximizes the margin between the nearest support vectors of both classes, using the chosen hyperplane (Yan & Zhu, 2022). In the KNN model, each data point is represented in an n-dimensional space, and when an unknown sample is introduced, the distance between the unknown sample and each data point is calculated based on the Euclidean distance (Alimjan et al., 2018).

Training the model

The parametrization of the models was done following the guidelines provided by Fonseca & Vieira (2023). All algorithms were executed using a cross-validation method with five-fold and 10 repetitions. The hyperparameter mtry, which determines the number of variables used as candidates at each split point, was fine-tuned using a random search with a tune length 10. The RF was performed with 500 trees, while the SGBoost was done with 250 and 500 trees. The models were evaluated using the total accuracy and kappa metrics (Vieira & Fonseca, 2022). Accuracy represents the ratio of correct responses to the total number of observations. Kappa statistics quantify the level of agreement between observed and expected values, taking into account the agreement that could occur by chance alone. Additionally, Kappa can be interpreted as the average reliability of categories or as an indicator of intraclass correlation (Warrens, 2015).

All the analyses were conducted in the iMESc—An Interactive Machine Learning App for Environmental Science, which is an open-source application built on R language (Vieira & Fonseca, 2022). Comprehensive details and step-by-step guidelines to extract the raw data are available at https://danilocvieira.github.io/iMESc_help/. The data can be accessed through “savepoint_acantholaimus” and “savepoint_sabatieria” in iMESC or in R following the same reference. The iMESc software can be downloaded at https://zenodo.org/record/7278042. The savepoints include both the datasets and the model’s results and outputs, which can be accessed by others for further analysis and validation. The save points ensure transparency and reproducibility of the study.

Results

Identification of Acantholaimus species

The accuracy of algorithms in identifying Acantholaimus species showed significant variability among them (Table 2). In the training of data, the RF algorithm achieved the highest accuracy of 94%, followed by SVM_L with 92% accuracy, and SVM_R with 92% accuracy (Table 2).

Table 2 Accuracies and Kappa index for the training and test part of the data from each algorithm used to construct the identification key: Random Forest (RF), Stochastic Gradient Boosting (SGboost), Support Vector Machine (SVM; linear (L) and radial (R)), and K-nearest neighbor (KNN). SD, standard deviations.

Models	Training	Testing	
	Accuracy	Kappa	Accuracy SD	Kappa SD	Accuracy	Kappa	
Acantholaimus							
RF	0.94	0.94	0.04	0.04	0.93	0.92	
SVM_L	0.92	0.91	0.05	0.05	0.92	0.92	
SVM_R	0.92	0.92	0.04	0.04	0.92	0.92	
SGboost	0.76	0.75	0.07	0.07	0.85	0.84	
KNN	0.51	0.49	0.06	0.06	0.78	0.76	
Sabatieria							
RF	0.97	0.97	0.02	0.02	1	1	
SVM_L	0.95	0.95	0.02	0.02	1	1	
SVM_R	0.93	0.92	0.03	0.03	0.97	0.96	
SGboost	0.74	0.73	0.04	0.04	0.90	0.90	
KNN	0.61	0.60	0.04	0.04	0.93	0.93	
Note:

Bold values indicate the highest accuracy and kappa index.

Upon evaluation of the testing dataset, the top four algorithms, including RF, SVM linear and radial, SGboost, and KNN, were able to accurately classify almost all specimens except for one individual of the species A. veitkoehlerae. (Id.47), which was misidentified as A. robustus (Table 3). When applied to the testing data, the RF algorithm yielded an overall accuracy of 93% and SVM; linear and radial achieved an accuracy of 92%, along with a corresponding kappa coefficient of 92% (Table 2).

Table 3 Percentages of accuracies (correct classifications), and errors (misclassifications) for each individual used to test the prediction of the Random Forest for Acantholaimus after calculating 500 trees.

Id, identification label of each individual; Species, species described in the original description; Predicted species, species predicted by the model.

Id	Accuracy (%)	Error (%)	Species	Predicted species	
Id.9	76	23	A.angustus	A.angustus	
Id.10	90	10	A.angustus	A.angustus	
Id.18	82	18	A.arthrochaeta	A.arthrochaeta	
Id.21	88	11	A.barbatus	A.barbatus	
Id.31	58	41	A.cornutus	A.cornutus	
Id.47	39	60	A.veitkoehlerae	A.robustus	
Id.52	81	18	A.sieglerae	A.sieglerae	
Id.64	96	4	A.veitkoehlerae	A.veitkoehlerae	
Id.65	98	2	A.veitkoehlerae	A.veitkoehlerae	
Id.74	70	29	A.quintus	A.quintus	
Id.81	78	22	A.verscheldi	A.verscheldi	
Id.89	66	33	A.microdontus	A.microdontus	
Id.99	30	69	A.septimus	A.septimus	
Id.108	41	58	A.megamphis	A.megamphis	
Note:

Bold value indicates the misclassified Acantholaimus species.

Out of the 29 characters analyzed, a subset of 17 characters stood out, comprising 8 morphometric measurements, seven quantitative ratios, and two categorical morphological characters (see Fig. 4). Several key characters emerged as highly significant across all models, including the diameter at the level of cephalic setae (DLCS), diameter at the level of cardia (DCL), body length without tail/length of the pharynx (b′), body length without tail (L′) and diameter at the level of the middle of the amphid (DLMA).

Figure 4 The Random Forest features importance analysis of the significant characters used in the identification of the Acantholaimus species.

The variables were ranked based on their average positions among the nodes of the 500 generated trees. The color gradient represents the position of the nodes (Minimal depth) in the trees. The higher the node position, the greater the variable importance. Abbreviations are listed in Table 1.

Identification of Sabatieria species

As for the Acantholaimus model, the algorithms with the Sabatieria species data showed significantly variable performance. Based on the training and testing data, the RF algorithm was the most accurate, followed by both SVM; linear and radial (Table 2).

Considering the testing part of the data, both RF and SVM (linear) models demonstrated a perfect global accuracy and kappa coefficient of 100%, whereas SVM (radial) achieved an accuracy of 97% and kappa of 96%. This success encompassed the accurate identification of all species (Table 4).

Table 4 Percentages of accuracies (correct classifications), and errors (misclassifications) for each individual used to test the prediction of the Random Forest for Sabatieria after calculating 500 trees. Id, identification label of each individual; Species, species described in the original description; Predicted species, species predicted by the model.

Id	Accuracy (%)	Error (%)	Species	Predicted species	
Id.3	73	27	S.alata	S.alata	
Id.8	74	26	S.armata	S.armata	
Id.13	67	33	S.balbutiens	S.balbutiens	
Id.28	88	13	S.celtica	S.celtica	
Id.30	90	10	S.celtica	S.celtica	
Id.36	78	23	S.conicauda	S.conicauda	
Id.44	97	3	S.conicoseta	S.conicoseta	
Id.59	86	14	S.elongata	S.elongata	
Id.64	82	18	S.execulta	S.execulta	
Id.76	22	77	S.fidelis	S.fidelis	
Id.81	87	12	S.granifer	S.granifer	
Id.88	63	37	S.granifer	S.granifer	
Id.108	100	0	S.lepida	S.lepida	
Id.112	100	0	S.lepida	S.lepida	
Id.115	94	6	S.longicaudata	S.longicaudata	
Id.121	97	2	S.longispinosa	S.longispinosa	
Id.145	74	26	S.multisupplementia	S.multisupplementia	
Id.153	100	0	S.ornata	S.ornata	
Id.158	100	0	S.ornata	S.ornata	
Id.166	69	31	S.parabyssalis	S.parabyssalis	
Id.169	54	46	S.parapraedatrix	S.parapraedatrix	
Id.180	58	42	S.pisinna	S.pisinna	
Id.183	94	7	S.pomarei	S.pomarei	
Id.190	35	65	S.praedatrix	S.praedatrix	
Id.195	96	4	S.propisinna	S.propisinna	
Id.206	100	0	S.pulchra	S.pulchra	
Id.216	100	0	S.pulchra	S.pulchra	
Id.222	96	4	S.punctata	S.punctata	
Id.226	100	0	S.punctata	S.punctata	
Id.232	62	38	S.sinica	S.sinica	
Id.242	82	18	S.stekhoveni	S.stekhoveni	
Id.246	95	4	S.stenocephalus	S.stenocephalus	
Id.258	80	19	S.vasicola	S.vasicola	

In the case of Sabatieria, the feature importance analysis selected a subset of 16 characters among the 30 used. Nine of them were morphometric measurements, four quantitative ratios, and three categorical morphological characters (Fig. 5). Notably, characters such as apophyses (Apoph), spicules (Spic), pharynx length (Pha. L), length of cephalic setae (CSL) and pharyngeal bulb body diameter (Pha. BBD) held prominent positions in the analysis, indicating their significance as the most important characters.

Figure 5 The Random Forest feature importance analysis of the significant characters used in the identification of the Sabatieria species.

The variables were ranked based on their average positions among the nodes of the 500 generated trees. The color gradient represents the position of the nodes (Minimal depth) in the trees. The higher the node position, the greater the variable importance. Abbreviations are listed in Table 1.

Discussion

The utilization of machine learning algorithms has demonstrated its effectiveness in identifying Acantholaimus and Sabatieria species. The findings that RF was the top-performing algorithm and KNN the least accurate agree with the literature (Liu et al., 2022). RF possesses the capability to handle large numbers of input variables and assign varying importance to each, thus effectively managing errors in datasets (Wäldchen & Mäder, 2018). RF also showed superior performance in the identification of wood species (Shugar, Drake & Kelley, 2021). KNN, in contrast, is known to be sensitive to outliers and becomes less efficient when dealing with large volumes of data (Cao et al., 2018). SVM also showed high accuracy values. This algorithm is normally applied to the classification of high-dimension data with many features, offering a fast classification process (Kremic & Subasi, 2016). The fact that RF performed better here does not mean that it will always outperform the others. Therefore, the recommendation is to compare the results of different algorithms, and eventually even an ensemble.

The construction of a comprehensive database of morphological characteristics is critical for implementing the proposed methodology across the phylum. In the case of the two genera studied here, the availability of outstanding systematic reviews (Jensen, 1979; Platt, 1985; Miljutin & Miljutina, 2016b) greatly facilitated the selection of relevant characteristics. While these reviews highlight several important characteristics for distinguishing species, not all of them were included in the analysis of this study. For instance, the complex structure of the copular apparatus (spicules and gubernaculum) and the shape of the buccal cavity in Acantholaimus were omitted from the analysis due to the challenging nature of categorizing them. The shape of the buccal cavity, in particular, is influenced by the degree of retraction/eversion of the stoma which is a result of the fixation method (Miljutin & Miljutina, 2016b). Similarly, the degree of eversion may also influence the head shape so this character must be used cautiously. In that case, however, we decided to keep the character since it was consistently present in individuals of each described species and was generally combined with other relevant morphological traits such as the length of cephalic setae and amphids’ position.

In the scope of this study, from the initial selection of 29 characters for Acantholaimus and 30 for Sabatieria, the feature importance analysis yielded a result of 17 (Acantholaimus) and 16 (Sabatieria) key characters for each genus. For Acantholaimus, significant features included morphological aspects such as amphid size and cervical setae position alongside specific morphometric attributes like the De man ratios. In the context of Sabatieria species, the analysis selected the characters related to the copular apparatus together with the tail length and the number of amphideal turns. In practice, if, these sets of characters are observed during the identification processes, it will enhance the chances of the model performing an accurate identification. On the other hand, a set of 12 and 14 characters for Acantholaimus and Sabatieria, respectively, were less relevant for distinguishing the species. Yet, the reasons why one character is more informative than another are a matter of further investigation. It could be that the selected characters have gone through disruptive selection (Rueffler et al., 2006) In that way, the implementation of a ML identification key facilitates the selection of the main traits to be used during the species identification process (Bogale, Baniya & DiGennaro, 2020; Tan et al., 2021), as well as gives us elements to further explore potential evolutionary process (Avila & Mullon, 2023).

The proposed approach does not eliminate the steps involved in the identification: observing the specimens, taking measurements, and categorizing the morphological characters. Instead, by leveraging the use of ML algorithms in taxonomy, it ensures a unified database and identification procedure for all researchers. As such, it allows the results of the identification processes to be equivalent across studies. Having comprehensive and well-documented species descriptions that cover multiple individuals and morphological characters is crucial for the success of the ML identification key. The more observations and detailed descriptions available, the better the quality and accuracy of the key. This issue is particularly important for species with strong sexual dimorphisms (Decraemer, Coomans & Baldwin, 2013). It is important to emphasize that the number of observations plays a central role in ML methods. Sufficient individuals are needed to train the models, and a separate set of individuals is required for testing and validation. Single individual descriptions pose challenges and limit the effectiveness of such methods, as they do not capture variation within a species. To implement this approach effectively, it would be advisable to start with taxonomic groups that have recent and comprehensive systematic reviews, such as Chromadoridae (Venekey et al., 2019) and Cyatholaimidae (Cunha, Fonseca & Amaral, 2022). These groups serve as the foundation for the morphological database and training of the ML models. As more comprehensive reviews become available for other taxonomic groups, the methodology can be extended to cover a wider range of marine nematode species.

It is important to acknowledge that misclassification can occur in ML algorithms, as observed for A. veitkoehlerae. The limited number of observations for certain morphological characters in this study may have contributed to the errors. ML algorithms rely on informative features extracted from the observations, which in this study are the specimens, to make accurate classifications (Bartlett et al., 2022). If the chosen features lack sufficient information or fail to capture the essential characteristics of the specimens, the algorithm performance will be compromised. Incorporating additional data, either new morphological characters or more individuals that capture the relevant variation within and among species, will enhance the algorithms’ predictive power. Thus, accurate taxonomic descriptions are crucial to achieve a better identification key.

There are, however, some limitations in implementing the tool for identifying Acantholaimus and Sabatieria species. The genus Acantholaimus benefits from having a significant number of described species, each based on detailed observations of four to seven individuals, with many of these species having been recently described. Conversely, Sabatieria poses challenges due to the descriptions being, in many cases, based on single or inadequately characterized individuals (Allgén, 1953; Wieser, 1954). Some descriptions focused only on females or males and there are instances where only (Micoletzky, 1924; Sergeeva, 1973) juveniles were included (Allgén, 1929). As a result, a considerable number of species (47 in total) could not be included in the analysis due to insufficient information and possessing somewhat incomplete descriptions. Future taxonomic efforts should prioritize obtaining multiple individuals at different life stages and sexes to address these limitations. The species left out from the analysis could be recollected and better described. Such an effort would provide a more robust identification tool covering a greater number of species. The ML identification key can be continuously improved and refined as more data (i.e., morphological characters, individuals, and species) becomes available, ensuring its accuracy and reliability in future applications.

Finally, when it comes to the identification of nematodes, it is of utmost importance to clarify the morphological characteristics and establish standardized terminology for these features. This ensures that researchers consistently use the same names to refer to the same structures (Decraemer, Coomans & Baldwin, 2013). A prime example is the case of supplements found in Sabatieria, which can exhibit pore-like or tubular appearances, essentially representing the same structure but describe with different terms (Leduc, 2013; Botelho et al., 2007; Botelho, Esteves & Fonsêca-Genevois, 2014). Such variations in terminology create confusion and hinder accurate identification. By promoting uniformity in character descriptions and adopting standardized terminology, we can greatly enhance the accuracy and clarity of nematode identification. This practice allows researchers to communicate effectively, compare findings across studies and build a comprehensive understanding of nematode anatomy (De Ley, 1995; Jenner, 2004).

Conclusion

This study showed that ML techniques can identify species of free-living marine nematodes. We suggest performing multiple algorithms to choose the most appropriate one. The results indicate that based on the presence/absence of morphological characters and a morphometric table, the process of identifying marine nematodes can be performed by algorithms, substituting the process of running traditional identification keys. Implementing ML keys can improve the speed and accuracy of species identification and allow researchers to efficiently process vast amounts of data. This approach also empowers non-taxonomists to confidently perform reliable identifications. Ultimately, introducing ML algorithms in taxonomy will contribute to our understanding and conservation of biodiversity. The success of having these keys depends on the quality of descriptions and systematic reviews.

Supplemental Information

Supplemental Information 1 List of valid Acantholaimus species according to Worms.

The green color indicates species that were excluded from the analysis due to poor taxonomical descriptions either by the absence of information of characters or were limited to a single specimen.

Click here for additional data file.

Supplemental Information 2 List of valid Sabatieria species according to Worms.

The green color indicates species that were excluded from the analysis due to poor taxonomical descriptions either by the absence of information of characters or were limited to a single specimen.

Click here for additional data file.

Supplemental Information 3 Number of individuals for Acantholaimus species Classification.

The number of individuals required for carrying out the classification of Acantholaimus species.

Click here for additional data file.

Supplemental Information 4 Number of individuals for Sabatieria species classification.

The number of individuals required for carrying out the classification of Sabatieria species.

Click here for additional data file.

Supplemental Information 5 Savepoint_acantholaimus.

Analytical details and all data.

Click here for additional data file.

Supplemental Information 6 Savepoint_sabatieria.

Analytical details and all data.

Click here for additional data file.

The authors extend their appreciation to Luciana Yaginuma, Nilvea Ramalho, Mauricio Shimabukuro, and Maikon Di Domenico for their invaluable support and contributions throughout the project. Additionally, the authors are also thankful to the dedicated members of the meiofauna team from UNIFESP and USP for their assistance and commitment to processing the samples. We would also like to express our gratitude to the reviewers, Dr. Jose Andrés Pérez-García and anonymous reviewers, for their comments, which significantly enhanced the quality of the manuscript.

Additional Information and Declarations

Competing Interests

Author Contributions

Data Availability

The authors declare that they have no competing interests.

Simone Brito de Jesus conceived and designed the experiments, performed the experiments, analyzed the data, prepared figures and/or tables, authored or reviewed drafts of the article, and approved the final draft.

Danilo Vieira analyzed the data, authored or reviewed drafts of the article, and approved the final draft.

Paula Gheller performed the experiments, prepared figures and/or tables, authored or reviewed drafts of the article, and approved the final draft.

Beatriz P. Cunha conceived and designed the experiments, prepared figures and/or tables, authored or reviewed drafts of the article, and approved the final draft.

Fabiane Gallucci conceived and designed the experiments, authored or reviewed drafts of the article, and approved the final draft.

Gustavo Fonseca conceived and designed the experiments, analyzed the data, authored or reviewed drafts of the article, and approved the final draft.

The following information was supplied regarding data availability:

The datasets and the model’s results and outputs are available in the Supplemental Files.

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
