# Peer review of "Machine learning algorithms accurately identify free-living marine nematode species"

_PeerJ, doi:10.7717/peerj.16216_

## Round 0.1 · original submission · Major Revisions

I agree with the reviewers that the manuscript requires major revision to improve its content and readability. In particular, reviewers agreed that the methodology section needs to be rewritten. Kindly address all concerns or suggestions of the reviewers in a point-to-point rebuttal and I look forward to reading the revised version of the manuscript.

Reviewer 1 ·

Basic reporting

This manuscript proposed a machine-learning approach to identify free-living marine nematode species. The topic is interesting and would be impactful in automating the identification process of nematode species. The write-up is clear and easy to follow in professional English.
However, there is no background study provided on the free-living marine nematodes, the importance of the species in the ecosystem & etc.
Similarly, there are no similar machine learning/deep learning nematode studies discussed in the introduction/background and what is the significance of the proposed study if compared to the literature studies.
For the raw data: “Further analytical details and the data are available in the supplementary material as save-points of the iMESC, specifically .savepoint_acantholaimus. and .savepoint_sabatieria.” Please provide the guidelines on how to extract the raw data and reproduce the results.

Experimental design

The methodology used particularly on the machine learning parts is ambiguous, a major rewrite is needed for this part, e.g.:
- It is not clear whether 2 separate identification models were built for each Acantholaimus and Sabatieria species.
- The identification was done at the genus level or species level?
- The detailed distributions or individuals for each species. What is the maximum and minimum number of individuals for each species?
- The software/tools used.
- Please justify this statement: “Only species of Acantholaimus and Sabatieria, which were described based on more than 4 individuals, were included in the split.” What is the purpose of including those data in the training, but not in the testing part?
- Why was it necessary to split the input data when implementing 5-fold cross-validation?
- The model could be overfitted if the number of samples/individuals was very small i.e. less than 30 individuals per species.

Validity of the findings

- What is meant by “error” in Tables 3 & 4?
- Why the sum of “accuracy” and “error” is 99 for most of the IDs in Tables 3&4?
- Should provide confusion matrix and AUC to better reflect the identification of the species.
- What is the aim of performing the feature importance analysis?
- In the discussion, the authors stated that “To implement this approach effectively, it would be advisable to start with taxonomic groups that have recent and comprehensive systematic reviews, such as Chromadoridae (Venekey et al., 2019) and Cyatholaimidae (Cunha et al., 2022). These groups serve as the foundation for the morphological database and training of the ML models.” Thus, it is wondered why these 2 genera of Acantholaimus and Sabatieria were selected for the proposed work.

·

Basic reporting

The manuscript is generally written in clear, unambiguous language. The subject that it addresses is current and is related to the recent impulse that the development of artificial intelligence and its applications in different branches of research have received. The references used are up to date and provide sufficient background, although some suggestions are made. The manuscript is well structured and the results are generally consistent with the proposed objectives.

Experimental design

The contribution to knowledge made by this research is clear. The description of the methodology allows the research to be replicated. Several suggestions are made to the authors regarding the way in which the materials and methods are described.

The raw data are provided by the authors; the use of an additional app to access them is pointed out. The conclusions of the paper are related to the proposed objective.

The main problems encountered are related to the scope of the proposed conclusions (lines 416-418; 418-419). I think the authors should adjust them better so that they can be justified by the results. See comments below.

There are also several details in the materials and methods section that I discuss further below.

Validity of the findings

no comment

Additional comments

Introduction
I recommend that the authors include in the paragraph between lines 63-73 the available bibliography on nematode research using algorithms (e.g. Bogale et al. 2020; Hakim et al. 2018 and Wang et al. 2020). This will allow to establish the state of the art of the topic. Likewise, I believe that the authors should emphasize the novelty of this type of approach to species identification. It is an exciting field that is just beginning now!


Material and methods
It would be advisable to explain what criteria were used to select these particular genera. If it is by number of species described, there are other more diverse genera such as Daptonema (167 spp) or Oncholaimus (129 spp) (Hodda 2022).

I agree that Acantholaimus spicules can be complex and difficult to analyze numerically. However, I think there is some ambiguity in the criteria for selecting characters. I explain: the same authors that discourage the use of spicules in the genus also warn that the shape of the head it is not a discrete character, it is subjective, and its appearance is dependent on the degree of retraction/eversion of the stoma (Miljutin & Miljutina 2016) Can you argue why you decided to keep this character? Another doubt: Was the influence of sex on morphological variables considered in any way? It is known that variables such as body length, tail length or amphid size among others can present sexual dimorphisms (Decraemer et al. 2014).

In lines 229-230 you state that the categorical morphological characters were transformed into numeric variables using two techniques: "Integer Encoding" and "One-Hot Encoding". Can you explain what criteria were followed in deciding which type of transformation is applied to each character? For example the character "cuticular ornamentation" is treated as binary in Acantholaimus, and as integer in Sabatieria. The need to use two types of transformations is also not clearly explained.

In the paragraph between lines 248-254 there is apparently a second reduction in the data set to be analyzed, as species with fewer than four described individuals are excluded. Is this the case?

You should check what is stated here: "If the species description was based on more than 10 individuals, two individuals were left out for validation purposes" (lines 250-251). This differs from the methodology presented in the abstract: "When more than 4 individuals have been described in the original publication, 25% were left out as test to validate the predictions of the model" (Lines 32-33). (Lines 32-33) Why is “four” chosen as the minimum acceptable number of individuals, why not three or five?

There is no explanation of the criteria used to exclude the two nematodes in cases where there were more than 10? Was it a random selection, were the last two to appear in the original description selected, was it a selection by size ....?

I believe that the number of species used in the validation should be included and not only the number of individuals

In lines 278-279 it is mentioned that the data and supplementary material are available as files generated by the app iMESC. Why is it necessary to use this app? According to its developers (https://zenodo.org/record/6484391) it would only allow to run the Random Forest algorithm. Wouldn't it be more convenient if the dataset was kept available in a more accessible format that does not require a third party app (e.g. svc)? I also recommend that the authors cite the program that was used to run the algorithms.

Results

I think that the Figures 3 and 4 are underutilized. They are mentioned in lines 308 and 327 but there is no proper discussion of these results. For example, what could be the explanation for the selection of that set of characters as the most relevant. What practical implications does it have that the set of most relevant characters varies in dependence on genus? This seems to me to contradict what you state in the Discussion (lines 355-356): "The set of characters selected by the analysis should be prioritized in routine identification and in future species descriptions".

Discussion

I consider that the scope of discussion can be further expanded on several points:
1. The authors could address the possible practical application taking into account that in a real sediment sample the variability of morphological characters may be greater than what is reported in the original descriptions.
2. Would it be possible to discriminate between genera using this type of analysis, and why was this possibility not tested in the present study?
3. I consider that one of the limitations of the analysis is the number of species that are left out, that is, if for Sabatieria I have to leave out 48 species, this means a reduction of the diversity of the taxa and the existence of a high possibility of assigning specimens to species that are not. In this scenario, how does the use of algorithms facilitate my work as a taxonomist?

The sentence in line 359 (Instead, it streamlines and enhances the identification process for marine nematodes) should be tempered. In the current state of art it does not seem to really facilitate the identification process. First I have to do all the processing of the samples until the mounting of the organisms (inevitable of course), but I have to identify the genera (which is already complex in some cases) and then make a table of morphological characters for each genus (with different characters) to run an analysis where a large number of species have been excluded. Is it really facilitating the work or is it even correct to use this type of shorcuts? This is a question that I have been wondering about throughout the manuscript.

Conclussions:
I do not consider that the sentence included in lines 416-418 is supported by the results obtained. I even consider that this should be discussed further by authors.

References:
Authors should review the guidelines, especially the use of et al. in references.

There are problems with several DOIs not working. They redirect to a personal folder (file:///E:/coisas%20artigo/DOI10.1109/MCI.2015.2471235).
By example:
Ren Y, Zhang L, Suganthan PN. 2016.Ensemble classification and regression-recent
developments, applications, and future directions. IEEE Computational intelligence magazine,
11(1): 41-53.DOI10.1109/MCI.2015.2471235.

Ridall A, Ingels J.2021. Suitability of free-living marine nematodes as bioindicators: Status and
future considerations. Frontiers in Marine Science, 8: 685327.DOI 10.3389/fmars.2021.685327.

Sagi O, Rokach L. 2018.Ensemble learning: A survey. Wiley Interdisciplinary Reviews: Data
Mining and Knowledge Discovery, 8(4):1249.DOI 10.1002/widm.1249.

The following references are not cited in the body of the text or have problems with the names:

Bisgin H et al., 2018.Comparing SVM and ANN based machine learning methods for species
identification of food contaminating beetles. Scientific reports, (8)1:1-12. DOI 10.1038/s41598-
018-24926-7.

De Bruyne K, Slabbinck B, Waegeman W, Vauterin P, De Baets B, Vandamme P. 2011.Bacterial species identification from MALDI-TOF mass spectra through data analysis and machine learning. Systematic and applied microbiology, 34 (1):20-29. DOI 10.1016/j.syapm.2010.11.003.

Gårdmark A. et al., 2013. Biological ensemble modeling to evaluate potential futures of living
marine resources. Ecological applications, 23(4):742-754.DOI 10.1890/12-0267.1.

González S, García S, Del Ser J, Rokach L, Herrera F. 2020. A practical tutorial on bagging
and boosting based ensembles for machine learning: Algorithms, software tools, performance
study, practical perspectives, and opportunities. Information Fusion, 64: 205-237.
DOI 10.1016/j.inffus.2020.07.007.

Grenouillet G, Buisson L, Casajus N, Lek S.2011.Ensemble modelling of species distribution:
the effects of geographical and environmental ranges. Ecography, 34 (1):9-17.
DOI 10.1111/j.1600-0587.2010.06152.x.

Guo Y, Chang Y, Yang P. 2018.Two new free-living nematode species (Comesomatidae) from
the mangrove wetlands in Fujian Province, China. Acta Oceanological Sinica, 37:161-167.
DOI 10.1007/s13131-018-1320-3.
In this case the authors cite Guo 2022.

Hao T, Elith J, Guillera0Arroita G, Lahoz0Monfort, JJ. 2019. A review of evidence about use
and performance of species distribution modelling ensembles like BIOMOD. Diversity and
Distributions, Diversity and Distributions, 25(5):839-852.DOI 10.1111/ddi.12892.

Muthumbi, A. W.; Soetaert, K; Vincx, M. Deep-sea nematodes from the Indian Ocean: new and
known species of the family Comesomatidae. Hydrobiologia, v. 346, n. 1, p. 25-57, 1997.

Ren Y, Zhang L, Suganthan PN. 2016.Ensemble classification and regression-recent
developments, applications, and future directions. IEEE Computational intelligence magazine,
11(1): 41-53.DOI10.1109/MCI.2015.2471235.

Sagi O, Rokach L. 2018.Ensemble learning: A survey. Wiley Interdisciplinary Reviews: Data
Mining and Knowledge Discovery, 8(4):1249.DOI 10.1002/widm.1249.

SandulIi R, Semprucci F, Balsamo M. 2014. Taxonomic and functional biodiversity variations
of meiobenthic and nematode assemblages across an extreme environment: a study case in a Blue
Hole cave. Italian Journal of Zoology, 81(4)508-516. DOI 10.1080/11250003.2014.952356.

Vanaverbeke, J. et al. The metazoan meiobenthos along the continental slope of the Goban Spur
(NE Atlantic). Journal of Sea Research, v. 38, n. 1-2, p. 93-107, 1997.

Vieira DC, Fonseca G. 2022. iMESc: An Interactive Machine Learning App for Environmental
Science (Version 2.1.0.1) Computer software. DOI 10.5281/zenodo.6484391. 2022.

Reviewer 3 ·

Basic reporting

Based on the manuscript, this study aims to evaluate several algorithms in machine learning to identify species using marine nematodes as case study. The title of the manuscript did not reflect the study. The word “accurately” in the title can be improved as there might be future studies that are required to validate the species accurately. Several comments have been provided as follows.

Experimental design

The methods can be organized into several subsections, especially for the data analysis part.

Validity of the findings

No comment.

Additional comments

Suggestions:
Summarize the study in the introduction
Methods can be reorganized into several subsections.
Data analysis can be organized into a few parts.
Adding a flowchart of methods might explain the overall methods.
Be consistent in writing the machine learning /ML throughout the manuscript
Some references are missing (e.g. Bartlett et al. 2022, Tumanov et al. 2020, etc.)

---

## Round 0.2 · accepted · Accept

I agree with the reviewers that the authors did a wonderful job in addressing the raised concerns and suggestions. The current manuscript is suitable for publication in PeerJ.

Reviewer 1 ·

Basic reporting

The authors have addressed my concerns and improved the quality of the manuscript.

Experimental design

No comment

Validity of the findings

No comment

Additional comments

Are the references in lines 301 and 305 the same? Fonseca & Vieira (2023) or Vieira & Fonseca, (2022)?

Reviewer 3 ·

Basic reporting

All comments have been addressed by the authors.

Experimental design

No comment

Validity of the findings

No comment

Additional comments

No comment